Comprehensive analysis of mitogen-activated protein kinase cascades in chrysanthemum

Song Aiping 1
Hu Yueheng 1
Ding Lian 1
Zhang Xue 1
Li Peiling 2
Liu Ye 1
Chen Fadi 1 chenfd@njau.edu.cn
1 College of Horticulture, Nanjing Agricultural University, Key Laboratory of Landscape Agriculture, Ministry of Agriculture , Nanjing, Jiangsu , China
2 College of Horticulture, Xinyang Agricultural and Forestry University , Xinyang, Henan , China
Uversky Vladimir
Electronic publication date: 2018 Jun 19
Publication date: 2018
Volume: 6
Electronic Location ID: e5037
Received 2018 Apr 24; Accepted 2018 May 30
Copyright: © 2018 Song et al.
Copyright year: 2018
Copyright holder: Song et al.
License: This is an open access article distributed under the terms of the Creative Commons Attribution License, which permits unrestricted use, distribution, reproduction and adaptation in any medium and for any purpose provided that it is properly attributed. For attribution, the original author(s), title, publication source (PeerJ) and either DOI or URL of the article must be cited.
License URL: https://creativecommons.org/licenses/by/4.0/

Keywords: Chrysanthemum morifolium, Mitogen-activated protein kinase, Phylogenetic analysis, Protein-protein interaction, Stress and development response

Funding: National Science Fund for Distinguished Young Scholars 31425022 National Natural Science Foundation of China 31730081, 31501792, 31700620 Natural Science Fund of Jiangsu Province BK20170722 333 High Level Personnel Training Project of Jiangsu Province BRA2017382 Foundation of Key Laboratory of Landscaping KF201801 Ministry of Agriculture, P.R. China This study was funded by the National Science Fund for Distinguished Young Scholars (31425022), the National Natural Science Foundation of China (31730081, 31501792, 31700620), the Natural Science Fund of Jiangsu Province (BK20170722), 333 High Level Personnel Training Project of Jiangsu Province (BRA2017382) and the Foundation of Key Laboratory of Landscaping (KF201801), Ministry of Agriculture, P.R. China. The funders had no role in study design, data collection and analysis, decision to publish, or preparation of the manuscript.

==============================
Background

Mitogen-activated protein kinase (MAPK) cascades, an important type of pathway in eukaryotic signaling networks, play a key role in plant defense responses, growth and development.

Methods

Phylogenetic analysis and conserved motif analysis of the MKK and MPK families in Arabidopsis thaliana, Helianthus annuus and Chrysanthemum morifolium classified MKK genes and MPK genes. qRT-PCR was used for the expression patterns of CmMPK and CmMKK genes, and yeast two-hybrid assay was applied to clear the interaction between CmMPKs and CmMKKs.

Results

We characterized six MKK genes and 11 MPK genes in chrysanthemum based on transcriptomic sequences and classified these genes into four groups. qRT-PCR analysis demonstrated that CmMKKs and CmMPKs exhibited various expression patterns in different organs of chrysanthemum and in response to abiotic stresses and phytohormone treatments. Furthermore, a yeast two-hybrid assay was applied to analyze the interaction between CmMKKs and CmMPKs and reveal the MAPK cascades in chrysanthemum.

Discussion

Our data led us to propose that CmMKK4-CmMPK13 and CmMKK2-CmMPK4 may be involved in regulating salt resistance and in the relationship between CmMKK9 and CmMPK6 and temperature stress.

Introduction

Mitogen-activated protein kinase (MAPK) cascades are a ubiquitous signal transduction pathway that is widely distributed in eukaryotes, transferring and amplifying external signals by the phosphorylation of downstream proteins (Hamel et al., 2006; Kazuya et al., 2002). The typical MAPK cascade pathway consists of MAPKKK-MAPKK-MAPK. MAPKKKs, as key molecules downstream of receptor-like protein kinases (RLKs), can be activated by protein phosphorylation (Wang et al., 2015). The MAPKs ultimately activated transduce extracellular environmental signals to cells by activating downstream response factors such as kinases, enzymes and transcription factors, regulating a series of intracellular reactions such as cell growth, differentiation, apoptosis and stress response (Ren et al., 2008).

MAPKK is a bispecific protein kinase that is activated by phosphorylation of the threonine and tyrosine residues of the TXY motif in the MAPK activation loop. According to the conserved S/T-xxxxx-S/T sequence and the D site, these motifs can be divided into Groups A, B, C and D (Kazuya et al., 2002). The MAPK protein family is a promiscuous family of serine/threonine kinases that phosphorylates a variety of substrates, including transcription factors, protein kinases and cytoskeleton-related proteins (Zhang et al., 2014).

Mitogen-activated protein kinase is reported not only to be widely involved in the transduction of plant stress signals in activating the expression of stress-resistance genes (Kong et al., 2013) but also to participate in signaling associated with plant growth and development (Xu & Zhang, 2015). OsMKK6-OsMPK3 is a signaling pathway that responds to cold stress in rice (Xie, Kato & Imai, 2012). The GhMKK4-GhMPK20-GhWRKY40 cascade in cotton plays important roles in the pathogenesis of Fusarium oxysporum (Wang et al., 2018). Plant growth and development require precise coordination among cells, tissues and organs. In eukaryotes, cell–cell and cell-environment communication usually involves cell surface receptors (Xu & Zhang, 2015). The MKK7-MPK6 signaling pathway regulates polar auxin transport to determine shoot branching in Arabidopsis thaliana (Jia et al., 2016). YODA and MPK6 were found to participate in embryonic root development in A. thaliana via growth regulation and cell division orientation (Smékalová et al., 2014). The YDA-MKK4/MKK5-MPK3/MPK6 cascade regulates local cell proliferation downstream of the endoplasmic reticulum (ER) receptor, which shapes plant organ morphology (Meng et al., 2012). In addition, the ER-YDA pathway regulates an immune surveillance system conferring broad-spectrum disease resistance (Sopeña-Torres et al., 2018).

Chrysanthemum (Chrysanthemum morifolium), one of the four most famous cut flowers in the world, is in great annual demand worldwide and is susceptible to various biotic and abiotic stresses (An et al., 2014). The MAPK cascade genes have been investigated in model plants, including Arabidopsis, rice and tomato, but rarely characterized in chrysanthemum. Here, we isolated six MKKs and 11 MPKs in chrysanthemum based on a set of transcriptomic data. We performed a comparative phylogenetic analysis of MPKs and MKKs in A. thaliana, Helianthus annuus and C. morifolium genes in silico and investigated the transcription pattern of CmMPKs and CmMKKs in different organs of chrysanthemum, as well as in response to various phytohormones and abiotic stresses, using qRT-PCR. A yeast two-hybrid assay was applied to analyze the interactions between CmMKKs and CmMPKs. We found that CmMKKs and CmMPKs participated widely in plant stress responses and plant growth, laying the foundation for future research on the function of CmMPKs and CmMKKs in C. morifolium.

Materials and Methods

Plant materials and growth conditions

Cuttings of the C. morifolium cultivar “Jinba” were obtained from the Chrysanthemum Germplasm Resource Preservation Center (Nanjing Agricultural University, Nanjing, China), then rooted in vermiculite without fertilizer in a greenhouse. After 14 days, the cuttings were transplanted to their corresponding growth substrates and then subjected to a range of stress and plant hormone treatments.

Database searches and sequencing of full-length CmMPK and CmMKK cDNAs

Arabidopsis MPK and MKK protein sequences were downloaded from The Arabidopsis Information Resource (TAIR) database and used as query sequences to identify CmMPK and CmMKK genes in chrysanthemum. Multiple alignments among the identified CmMPK and CmMKK sequences were also performed to avoid repetition. Finally, seventeen pairs of gene-specific primers (Table S1) were designed to amplify the full open reading frame (ORF) sequences. The related amplicons were purified using an AxyPrep DNA Gel Extraction Kit (Axygen, Hangzhou, China) and ligated into pMD19-T (Takara, Tokyo, Japan) for sequencing.

Phylogenetic tree construction and sequence analysis

A phylogenetic tree was constructed with MEGA version 7.0 using the Maximum Likelihood method (Kumar, Stecher & Tamura, 2016). Multisequence alignments of CmMPK and CmMKK protein sequences were performed among A. thaliana, H. annuus (Badouin et al., 2017) and C. morifolium using MUSCLE (Edgar, 2004). The theoretical isoelectric point (PI) and molecular weight (MW) of CmMPK and CmMKK proteins were calculated using the Compute PI/MW online tool (http://web.expasy.org/compute_pi/), and PSORT online tool was used to predict their subcellular localization. The MEME v4.10.2 program (Bailey et al., 2015) was employed to identify the motifs present in the CmMPK and CmMKK proteins.

Plant treatments

The transcription profiles of CmMPK and CmMKK genes in roots, stems, unexpanded leaves, mature leaves and senescent leaves were explored. Various abiotic stresses were imposed, including high salinity (200 mM NaCl) and osmotic stress (20% w/v polyethylene glycol 6000, PEG 6000) (Song et al., 2012). For the NaCl and PEG 6000 assays, young plants were transferred to liquid medium containing the stress agent, and the second true leaves were sampled at various time points (Song et al., 2014a). Other seedlings were subjected to a period at either 4 °C or 40 °C in a chamber with 50 μmol·m−2·s−1 of light, the second true leaves were sampled after 1 h treatment (Song et al., 2014c). The phytohormone treatments involved spraying the leaves with either 50 μM abscisic acid (ABA), or 5 mM gibberellins (GA) (Song et al., 2014b). The plants were sampled at 0 h and after 1 h treatments. After sampling, all of the collected material was snap frozen immediately in liquid nitrogen and stored at −70 °C before RNA extraction. Three biological replicates per experiment were performed.

Real-time quantitative PCR

Total RNA was isolated from samples using the Quick RNA isolation Kit (Huayueyang, Beijing, China), with the RNase-free DNase I treatment to remove potential genomic DNA contamination. The first cDNA strand was synthesized from 1 μg of total RNA using PrimeScript™ RT reagent Kit with gDNA Eraser (Takara, Tokyo, Japan) according to the manufacturer’s instructions. qPCR was performed using a Mastercycler ep realplex instrument (Eppendorf, Hamburg, Germany). The qPCR reaction cocktail and cycling regime were applied as described by Song et al. (2016). Gene-specific primers (sequences shown in Table S1) were designed using Primer Premier 5, and the EF1α gene was employed as a reference. Relative transcript abundances were calculated via the 2−ΔΔCT method (Livak & Schmittgen, 2001). A total of three independent experiments were conducted.

Data analysis

For expression pattern analysis, the relative transcript expression levels of each CmMPK and CmMKK were log2 transformed, and the profiles were compared using Cluster v3.0 software (Hoon et al., 2011) and visualized using Treeview (Eisen et al., 1998). A one-way analysis of variance, followed by the use of LSD test (P = 0.05), was employed to statistical analysis.

Yeast two-hybrid assay

For the yeast two-hybrid assay, the CmMKK ORF fragments were cloned into the pGADT7 vector in-frame with the GAL4 activation domain (primers given in Table S1). The CmMPK ORF cDNA fragments were cloned into the pGBKT7 vector in-frame and proximal to the binding domain (primers refer to Table S1). The pairs of pGADT7-CmMKK and pGBKT7-CmMPK vectors were cotransformed into the Y2H yeast strain using the Matchmaker Gold Yeast Two-Hybrid System (Clontech, Mountain View, CA, USA). Positive clones were plated onto selective SD/-Leu/-Trp medium. A total of 2 days later, monoclones were picked and suspended in 200 μL of water, and 4 μL of the suspension was placed on SD/-Leu/-Trp/-His/-Ade medium and on SD/-Leu/-Trp/-His/-Ade/X-a-gal medium. The results could be observed after 2 days. Three independent experiments were conducted.

Results

Phylogenetic relationships among MPK and MKK proteins of A. thaliana, H. annuus and C. morifolium

Details regarding the isolated 11 CmMPK (GenBank: MG334201–MG334212) and six CmMKK (GenBank: MG334196–MG334201) sequences are provided in Table 1. The CmMKK and CmMPK proteins were predicted to have different subcellular localizations, including the chloroplast, cytoskeleton, cytoplasm, mitochondrion, or nucleus, based on PSORT analysis.

Table 1 Summary of CmMPK/CmMKK sequences and the identity of likely A. thaliana homologs.

Gene	GenBank accession no.	Amino acids length (aa)	AtMKK orthologs	Locus name	PI	MW	Subcellular loclization	
CmMKK2	MG334196	354	AT4G29810.1	MAP kinase kinase 2	5.47	38,986.44	cyto	
CmMKK3	MG334197	517	AT5G40440.4	Mitogen-activated protein kinase kinase 3	4.88	57,585.41	cyto, nucl	
CmMKK4	MG334198	360	AT3G21220.2	MAP kinase kinase 5	9.37	40,537.26	chlo, nucl	
CmMKK5	MG334199	371	AT3G21220.2	MAP kinase kinase 5	9.24	41,049.59	chlo, mito, nucl	
CmMKK6	MG334200	334	AT5G56580.1	MAP kinase kinase 6	6.37	37,432.26	nucl, cyto	
CmMKK9	MG334201	312	AT1G73500.1	MAP kinase kinase 9	8.03	35,245.67	cyto, nucl	
CmMPK1	MG334202	379	AT1G10210.3	Mitogen-activated protein kinase 1	6.64	43,816.73	cyto, cysk	
CmMPK3.1	MG334203	371	AT3G45640.1	Mitogen-activated protein kinase 3	5.39	42,568.67	cysk, cyto, nucl	
CmMPK3.2	MG334204	371	AT3G45640.1	Mitogen-activated protein kinase 3	5.34	42,670.78	cysk, cyto, nucl	
CmMPK4.1	MG334205	379	AT4G01370.1	MAP kinase 4	6.13	43,628.62	cyto, cysk	
CmMPK4.2	MG334206	387	AT4G01370.1	MAP kinase 4	6.28	44,200.39	cysk, chlo, nucl, cyto	
CmMPK6	MG334207	391	AT2G43790.1	MAP kinase 6	5.44	45,151.47	nucl, cyto	
CmMPK9.1	MG334208	507	AT3G18040.4	MAP kinase 9	6.43	57,992.35	cyto, cysk, nucl, chlo	
CmMPK9.2	MG334209	538	AT3G18040.3	MAP kinase 9	6.03	61,502.68	cyto, chlo, nucl	
CmMPK13	MG334210	381	AT1G07880.2	Protein kinase superfamily protein	5.81	43,434.13	chlo, cyto, nucl	
CmMPK16	MG334211	563	AT5G19010.1	mitogen-activated protein kinase 16	9.13	63,910.22	cyto, chlo, nucl	
CmMPK18	MG334212	599	AT1G53510.1	mitogen-activated protein kinase 18	9.41	68,021.07	cyto, nucl	
Note:

PI, isoelectric point; MW, molecular weight; chlo, chloroplast; cysk, cytoskeleton; cyto, cytoplasm; mito, mitochondrion; nucl, nucleus.

The phylogenetic tree constructed for the MPK protein family in A. thaliana, H. annuus and C. morifolium could be divided into four groups: A, B, C and D (Fig. 1). The motifs of Group A were similar to those of Group B, while motif 10 was unique to Group D. Group C possessed the fewest motifs and lacked motif 7, which was present in the other groups. The major members of Group A were MPK3/6/10. MPK3 had two paralogs in C. morifolium and H. annuus, whereas only one ortholog was found in A. thaliana, which had another member, MPK10, in this group. The major members of Group B were MPK4/5/11/12/13. MPK4 was similar to MPK3, and motif 9 was missing from H. annuus HanXRQChr04g0121371 in this group, while MPK5/11/12 was unique to Arabidopsis. The major members of Group C were MPK1/2/7/14 in A. thaliana, but C. morifolium had only one ortholog of MPK1, while H. annuus had three MPK1/2 homologs. In Group D, C. morifolium MPK9.1 was separate from MPK9.2, and MPK18/19/20 was found in both A. thaliana and H. annuus, while only MPK18 was found in C. morifolium. Only one copy of MPK16 was present in A. thaliana and C. morifolium, while two members could be found in H. annuus.

Figure 1 Phylogenetic and domain analyses of MPKs.

(A) Phylogenetic relationships of A. thaliana, H. annuus and C. morifolium MAPK. The phylogenetic tree was constructed by MUSCLE using the MEGA7.0 program. (B) Schematic diagram of the amino acid motifs of A. thaliana, H. annuus and C. morifolium MPKs. Motif analysis was performed using MEME 4.0 software as described in the methods. The black solid line represents the corresponding MPK and its length. The different-colored boxes represent different motifs and their positions in each MPK sequence. The blue circle: A. thaliana; the green square: H. annuus; the yellow triangle: C. morifolium.

The phylogenetic tree of the family of MKK proteins in A. thaliana, H. annuus and C. morifolium could also be divided into four groups: Groups A, B, C and D (Fig. 2). Group A contained motif 10, which could not be found in any other three groups, although the other three groups contained similar motifs. The major members of the Group A were MKK6/1/2, while MKK6 was the only ortholog in A. thaliana and C. morifolium, and H. annuus had two paralogs. Group B contained only MKK3, and there was only one ortholog in the species. The main members of Group C were MKK4 and MKK5, but one more member, HanXRQChr12g0360031, was present in H. annuus, which instead was missing motif 9 and motif 3 from this group. The major members of Group D were MKK7/8/9/10. Only MKK9 was found in H. annuus and C. morifolium, while MKK7/8/10 were unique to A. thaliana, and MKK10 in A. thaliana was lacking motif 5.

Figure 2 Phylogenetic and domain analyses of MKKs.

(A) Phylogenetic relationships of A. thaliana, H. annuus and C. morifolium MKKs. The phylogenetic tree was constructed by MUSCLE using the MEGA7.0 program. (B) Schematic diagram of amino acid motifs of A. thaliana, H. annuus and C. morifolium MKKs. Motif analysis was performed using MEME 4.0 software as described in the methods. The black solid line represents the corresponding MKK and its length. The different-colored boxes represent different motifs and their positions in each MKK sequence. The blue circle: A. thaliana; the green square: H. annuus; the yellow triangle: C. morifolium.

Differential expression patterns of CmMPK and CmMKK genes in different organs

The expression patterns of CmMPKs in various organs, including unexpanded leaves, mature leaves, senescent leaves, stems and roots (Fig. 3A), were quantitatively analyzed. The results showed that compared to those in unexpanded leaves, the expression levels of CmMPK1, CmMPK3.1, CmMPK4.1, CmMPK4.2, CmMPK6 and CmMPK16 in mature leaves and senescent leaves were significantly increased (Fig. 3A). Some pairs of paralogs, such as CmMPK3.1 and CmMPK3.2, CmMPK4.1 and CmMPK4.2, and CmMPK9.1 and CmMPK9.2, had different expression levels and expression trends in different organs. The expression levels of CmMPK3.2, CmMPK9.2 and CmMPK13 in the whole plant were lower than those of other CmMPKs. Quantitative analysis of CmMKK expression levels in different organ showed that the expression of CmMKK9 was significantly increased in mature and senescence leaves, while CmMKK4 was highest in the roots (Fig. 3B).

Figure 3 Expression patterns of CmMPKs (A) and CmMKKs (B) in different organs/tissues obtained by qRT-PCR analysis.

L1: unexpanded leaves; L2: mature leaves; L3: senescent leaves; S: stems; R: roots. All samples were run in triplicate, and the data were normalized relative to EF1a.

Differential responses of the CmMPKs and CmMKKs to abiotic stress

After heat shock and cold treatment for 1 h, the expression levels of CmMPKs were quantitatively analyzed (Fig. 4A). As shown, we found that CmMPK3.1, CmMPK3.2 and CmMPK4.2 were induced by cold treatment and did not respond to heat shock. CmMPK1, CmMPK9.1, CmMPK9.2, CmMPK16 and CmMPK18 were all induced by cold treatment, and their expression levels decreased after heat shock treatment. CmMPK4.1 was induced after heat shock treatment but did not respond to cold stress. CmMPK6 and CmMPK13 responded to cold and heat shock treatment, but the expression levels after heat shock treatment were slightly lower than those after cold treatment.

Figure 4 Expression patterns of CmMPKs (A) and CmMKKs (B) under cold and heat shock treatment in chrysanthemum obtained by qRT-PCR analysis.

CK: control plants; C4: the plants treated with 4 °C; H40: the plants treated with 40 °C. Details of the treatments are reported in “Materials and Methods”. All samples were run in triplicate, and the data were normalized relative to EF1a.

After heat shock and cold treatment for 1 h, the expression levels of CmMKKs were quantitatively analyzed (Fig. 4B). We found that CmMKK2, CmMKK3 and CmMKK5 all showed increased expression in response to cold, while their expression levels decreased after heat shock treatment. CmMKK9 responded to both cold and heat shock treatment, but the expression level after heat shock treatment was slightly lower than that after cold treatment. CmMKK4 and CmMKK6 did not respond to cold heat shock treatment.

The expression patterns of 11 CmMPKs and six CmMKKs after two abiotic stresses and two exogenous hormone treatments were analyzed by qRT-PCR (Fig. 5). The results showed that CmMPK3.2, CmMPK13 and CmMKK4 were induced after NaCl treatment, while the expression levels of CmMPK9.2 and CmMPK16 were significantly decreased. The expression of CmMPK18 increased significantly after ABA treatment, while the expression of CmMKK6 decreased significantly. CmMPK4.1 and CmMPK4.2 had similar expression patterns, but the expression patterns of two other pairs of paralogs, CmMPK3.1 and CmMPK3.2 and CmMPK9.1 and CmMPK9.2, showed differences. The expression patterns of CmMKK2 and CmMPK4.1 were similar, as were those of CmMKK5/CmMPK9.1, CmMKK4/CmMPK13 and CmMKK6/CmMPK16.

Figure 5 Expression patterns of CmMPKs and CmMKKs under abiotic stress treatments in chrysanthemum obtained by qRT-PCR analysis, and shown as a heatmap.

The details of the treatments are reported in “Materials and Methods”. All samples were run in triplicate, and the data were normalized relative to EF1a.

The interaction between CmMPKs and CmMKKs in the yeast two-hybrid assay

The yeast two-hybrid assay was applied to discover the interactions between CmMPKs and CmMKKs. As shown in Table S2 and Fig. 6, all CmMKKs could interact with CmMPK1/3/6. CmMPK4 interacted strongly with CmMKK2 and weakly with CmMKK4/5. CmMPK9/16 did not interact with any CmMKKs, CmMPK13 interacted weakly with CmMKK2/4/5/6, and CmMPK18 interacted only with CmMKK2.

Figure 6 Interactions of CmMKKs with CmMPKs in yeast.

The CmMKK ORF fragments were cloned into the pGADT7 (AD) vector in-frame with the GAL4 activation domain, while the CmMPK ORF cDNA fragments were cloned into the pGBKT7 (BD) vector in-frame with the GAL4 binding domain. pGADT7-T-CmMKKx was used as bait, and pGBKT7-CmMPKs were used as prey; (A) pGADT7-CmMKK2; (B) pGADT7-CmMKK3; (C) pGADT7-CmMKK4; (D) pGADT7-CmMKK5; (E) pGADT7-CmMKK6; (F) pGADT7-CmMKK9. In addition, pGADT7-T/pGBKT7-53 was used as a positive control, while pGADT7-T/pGBKT7-Lam was used as a negative control.

Discussion

Comparative analysis of the A. thaliana, H. annuus and C. morifolium MPK and MKK gene families

In this study, 11 CmMPK and six CmMKK genes were isolated in chrysanthemum based on transcriptome data and classified according to their sequence similarity. To elucidate the phylogenetic relationships among the CmMPK genes and to infer the evolutionary history of the gene family, we constructed a phylogenetic tree with high support rates using MPK members from A. thaliana, H. annuus and C. morifolium. The MPKs were well divided into four groups, and we found that most of the motifs of MPKs in the same group were similar to each other. MPK3 has been reported to play roles in abiotic resistance (Beckers et al., 2009) (Li et al., 2017), ovule development (Wang et al., 2008) and anther development (Hord et al., 2008). Only one MPK3 was found in A. thaliana, but two paralogs were found in both H. annuus and C. morifolium (Fig. 1). Additionally, the expression patterns of CmMPK3.1 and CmMPK3.2 were significantly different in different plant organs (Fig. 3A) and in response to temperature stresses (Fig. 4A) and other abiotic stresses (Fig. 5). This difference suggested that MPK3 was duplicated and that functional differentiation occurred after the speciation event that divided A. thaliana and the other two species. The same situation was also found in MPK4 (Group B) and MPK9 (Group D). MPK4 was reported to be required for light-induced anthocyanin accumulation (Li et al., 2016) and to be associated with cold resistance (Du et al., 2017). Notably compared to the other members of MPK4, one of the MPK4 proteins in H. annuus missed motif 9, and the functions of the two MPK4 paralogs in H. annuus were suspected to be differentiated. MPK9 was reported to be involved in stomatal closure in A. thaliana (Khokon et al., 2017). There were significant differences in the expression patterns between CmMPK9.1 and CmMPK9.2 in various organs of C. morifolium (Fig. 3A) and in response to temperature stress (Fig. 4A) and other abiotic stress treatments (Fig. 5). This evidence suggested that the functions of the two MPK9 paralogs were differentiated in H. annuus and C. morifolium. MPK3 and MPK6 were considered to be closely related and functionally redundant in A. thaliana, and they were also well grouped. H. annuus had one more MPK6 than A. thaliana, indicating that MPK6 was duplicated after the speciation event between A. thaliana and H. annuus. In Group A, AtMPK11 and AtMPK13 lacked motif 7, which was found in the other members, possibly implying that these two members underwent changes in function relative to other members of Group A. MPK13 in Group B also appeared to show similar behavior to MPK6. MPK10 in Group A, MPK11/5/12 in Group B and MPK17 in Group D appeared only in A. thaliana, and the function of these genes has rarely been reported in plants, possibly suggesting that the gene was functionally redundant and was lost during evolutionary history.

To elucidate the phylogenetic relationship among the MKKs and infer the evolutionary history of the MKK family, we constructed a phylogenetic tree with high support rates using MKK members from A. thaliana, H. annuus and C. morifolium. The members divided well into four groups. There were only two paralogs of C. morifolium and A. thaliana in Group C, while H. annuus contained an additional gene, HanXRQChr12g0360031, that lacked motifs 5/8/3, suggesting that pseudogenization may have occurred. In addition, we found that there was a large-scale expansion of Group D in A. thaliana, while C. morifolium and H. annuus each retained one ortholog, and the motifs in A. thaliana were also differentiated, suggesting that they were more important for survival in A. thaliana.

CmMPKs and CmMKKs are involved in plant growth and development

Many members of the MAPK cascade had been shown to participate in plant vegetative and reproductive growth (Xu & Zhang, 2015). In this study, we observed that the expression patterns of CmMPK1, CmMPK3.1, CmMPK4.1, CmMPK4.2 and CmMPK6 were similar in all organs of C. morifolium (Fig. 3A). With the maturation and senescence of leaves, the expression levels of those CmMPKs increased, while their expression in the root and stem remained at a relatively low level. These members were speculated to function in plant development. These CmMPKs interact with CmMKK2 and CmMKK4 in the yeast two-hybridization assay, and CmMPK3.1/6 interact with CmMKK9 (Fig. 6). Meanwhile, the expression patterns of CmMKK9 and CmMPK3.1/6 were similar (Fig. 3B). CmMKK9 was hypothesized to regulate plant growth by the phosphorylation of CmMPK3.1, CmMPK4.1 and CmMPK6. However, the expression levels of CmMKK2 and CmMKK4 during development differed from those of CmMPK3.1, CmMPK4.1 and CmMPK6. The expression of CmMKK4 and CmMKK6 in various organs instead appeared constitutive (Fig. 3B). These proteins were speculated to provides important dose-effects in the MAPK cascade. Earlier studies had shown that MKK9 could activate MPK3/6 and that the activation of the CmMKK9-MPK3/6 cascade induced ethylene synthesis (Xu et al., 2008), which was consistent with the results of the yeast two-hybrid assay in this experiment. Another MAPK cascade of A. thaliana, composed of MKK4/5 and MPK3/6, was shown to promote local cell proliferation by promoting downstream RLKs to regulate inflorescence structures (Meng et al., 2012). Leaf senescence was delayed in plants with single mutations of MKK9 or MPK6, and the overexpression of MKK9 led to premature leaf senescence. This evidence demonstrated that the MKK9-MPK6 cascade was involved in the regulation of leaf senescence (Zhou et al., 2009). According to their expression patterns in C. morifolium, CmMPK1 and these CmMPKs and CmMKKs, which had been previously shown to play a role in plant development, showed similar expression trends during plant development. CmMKK2 interacts with CmMPK3.1, CmMPK6 and CmMPK4.1 in the yeast two-hybrid system (Fig. 6) and had been shown to be upstream MKK of AtMPK4/6 in A. thaliana (Teige et al., 2004). This report was consistent with the results in this experiment, and CmMKK2 was also speculated to be involved in the regulation of plant development via the regulation of downstream MPKs.

CmMPKs and CmMKKs were involved in response to temperature stresses

Most members of the MAPK cascade had been reported to show a response to temperature stress. The MAPK cascade pathway in Brachypodium distachyon was temperature sensitive: 90% of the MAPK cascade kinase genes were induced under cold stress, and 60% of the genes were induced by high temperature stress (Jiang et al., 2015). Most Cucumis sativus CsMPKs (except CsMPK3, CsMPK7 and CsMPK13) were downregulated after cold treatment; however, most CsMAPKs were upregulated under heat shock stress except for CsMPK3 and CsMPK7 (Wang et al., 2015). The A. thaliana AtMEKK1-AtMAPKK2-AtMAPK4/AtMAPK6 pathway had been shown to play an important role in the defense against salt stress and cold stress (Teige et al., 2004). In contrast to Cucumis sativus (Wang et al., 2015), all CmMPKs except CmMPK4.1 in C. morifolium were induced after cold treatment (Fig. 4A), but the expression level of MPKs, except CmMPK4.1, CmMPK6 and CmMPK13, decreased or remained unchanged after heat shock treatment for 1 h (Fig. 4A), and it was hypothesized that the function of MPK differs among species. Experiments had shown that under the same stress conditions, some orthologs in C. sativus, A. thaliana, and Oryza sativa exhibit completely different expression patterns. For example, AtMPK7 was significantly upregulated and CsMPK7 was significantly downregulated under cold stress conditions. In addition, OsMKK4 was upregulated under drought and cold stress conditions, while CsMKK4 was downregulated under the same stress conditions (Wang et al., 2015). It was worth noting that the CmMKK9-CmMPK6 cascade had a consistent expression trend after temperature stress: CmMKK9 (Fig. 4A) and CmMPK6 (Fig. 4B) after cold and heat shock treatments, and we could surmise that this cascade may also be involved in the response to temperature stresses. It was noteworthy that only CmMPK4.1 was increased in expression after heat-treatment for 1 h but not cold-induced, whereas its paralog, CmMPK4.2, was in the opposite. In A. thaliana, AtMPK4 was reported to be a downstream component of H2S-associated cold stress, and both H2S and MPK4 response to cold stress by regulate cold response genes and stomatal movement (Du et al., 2017). CmMPK4.2 may play a similar role of AtMPK4 in the cold stress. To date, there had been no report on the response of MPK4 to heat shock stress. We speculate that two CmMPK4 paralogs were functionally differentiated and participated in response to heat shock stress or cold stress in plants, respectively.

CmMPKs and CmMKKs are involved in resistance to salt and osmotic stresses

Figure 3B shows that CmMKK4 was maintained at a low and stable level in most tissues but specifically expressed in the root, and its expression was significantly increased after PEG and salt treatments (Fig. 5). Additionally, the overexpression of ZmMKK4 in A. thaliana increased the tolerance to cold and salt stress relative to the control group by increasing the germination rate, lateral root number, plant viability, proline and soluble sugar content, chlorophyll and antioxidant enzyme activity (Kong et al., 2011). These facts suggest that CmMKK4 is involved in the response to salt stress. Both CmMPK13 and CmMKK4 were induced by salt and osmotic stresses, and they were also shown to interact weakly in the yeast two-hybrid assay (Fig. 6). We could speculate that CmMPK13 may be downstream of CmMKK4 in the MAPK cascade and play a role in salt and osmotic stress. This MEKK1-MKK1/MKK2-MPK4 cascade had previously been shown to work together in experiments on the regulation of plant innate immunity (Gao et al., 2008; Kong et al., 2012) in addition to reactive oxygen signals and salicylic acid signals (Pitzschke et al., 2009). The expression patterns of CmMPK4.2 and CmMKK2 of this cascade in C. morifolium were similar (Fig. 5), and they were also shown to interact strongly in yeast (Fig. 6). The expression levels of CmMPK4.2 and CmMKK2 increased after salt stress and PEG treatment. Under salt stress or other abiotic stress, the dynamic balance of active oxygen is disturbed, and membrane lipid peroxidation or membrane lipid degreasing leads directly to increased active oxygen content in plants (Oukarroum et al., 2015). According to the above evidence, we surmise that the MKK2-MPK4 cascade is activated under salt stress in C. morifolium and regulates the reactive oxygen system to combat abiotic stress. The expression patterns of CmMPK3.1 and CmMPK3.2 were quite different from each other (Fig. 5). CmMPK3.1 was not induced under abiotic stress treatment and exogenous hormone treatment, while the expression level of CmMPK3.2 significantly increased after salt treatment. In addition, as shown in Fig. 3, CmMPK3.1 expression in mature leaves and senescent leaves was significantly higher than in unexpanded leaves, while the expression of CmMPK3.2 in every tissue remained at a very low level with no obvious features. Therefore, we surmised that the two MPK3 paralogs in C. morifolium have functional differentiation: CmMPK3.1 plays a role in plant development, while CmMPK3.2 is involved in the salt stress response.

CmMPKs and CmMKKs are involved in the resistance to exogenous phytohormone treatments

The relationship between MAPK cascades and ABA signaling had been previously reported (Danquah et al., 2014; De Zelicourt, Colcombet & Hirt, 2016). MPK9 and MPK12 had been reported to act as downstream genes of ROS to positively regulate guard cell ABA signaling (Jammes et al., 2009). The MAP3K17/18-MKK3-MPK1/2/7/14 cascade had been shown to be an intact ABA-activated MAPK cascade (Danquah et al., 2015). In this study, the CmMPKs and CmMKKs were quantitatively analyzed in GA- and ABA-treated C. morifolium (Fig. 5). The results revealed that the expression of CmMPK1 did not change under exogenous GA or ABA treatment; the expression of CmMKK3, CmMPK9.1 and CmMPK9.2 increased slightly after exogenous ABA treatment; and the expression of CmMPK18 increased most obviously after ABA treatment. MPK18 has been reported in relation to vascular-related functions in A. thaliana (Benhamman et al., 2017). Whether MPK18 participates in the hormone response in C. morifolium needs further study.

The interactions between CmMPKs and CmMKKs revealed by the yeast two-hybrid assay

In this study, the relationship between CmMPK and CmMKK in C. morifolium was examined by a yeast two-hybrid assay. CmMKK2 could interact strongly with CmMPK4/6 in the yeast two-hybrid system (Fig. 6), which was consistent with the composite cascade of AtMEKK1-AtMKK2-AtMPK4/AtMPK6 that had been demonstrated in A. thaliana (Teige et al., 2004). The yeast two-hybrid assay was also used in A. thaliana to gain insight into the potential relationships between all A. thaliana AtMPKs and their upstream activator AtMKKs (Lee et al., 2008): AtMKK2 interacts with AtMPK4, AtMPK6, AtMPK10, AtMPK11 and AtMPK13 in A. thaliana. In contrast to A. thaliana, CmMKK2 also interacts with CmMPK3 and interacts weakly with CmMPK1/13/18. AtMKK4 interacts strongly only with MPK3/6 in A. thaliana (Lee et al., 2008), whereas CmMKK4 exhibits a stronger yeast two-hybrid interaction than AtMKK4, and CmMKK4 not only strongly interacts with CmMPK3/6 but also shows a weak interaction with CmMPK1/4/13. In addition, AtMKK5, a paralogous MKK of AtMKK4, was the upstream gene of AtMPK3 and AtMPK6 in the MAPK cascade, and the AtMKK4/5-AtMPK3/6 cascade plays a key role in the response to many different stresses (Asai et al., 2002). Unlike AtMKK4, AtMKK5 interacts only with AtMPK6 in the yeast system (Lee et al., 2008). However, CmMKK5 not only interacts strongly with CmMPK6 but also interacts with CmMPK3 and interacts weak interaction with CmMPK1/4/13 (Fig. 6). Interestingly, this phenomenon was not limited to MKK5. AtMKK9 does not interact with AtMPK6 in yeast two-hybrid assays, but it can actively phosphorylate AtMPK6 and AtMPK20 in vitro (Lee et al., 2008), and CmMKK9 not only interacts strongly with CmMPK3/6 in yeast two-hybrid experiments but also interacts weakly with CmMPK1 (Fig. 6). This discrepancy may be due to the limitations of the yeast system.

In the yeast two-hybrid assay, the results of the interactions between MKKs and MPKs in C. morifolium were different from those in A. thaliana. First, we speculated that the functions of MPKs in C. morifolium and their orthologs in A. thaliana had been differentiated. Second, false-positive and false-negative interference occurred in the yeast two-hybrid assay. In the yeast two-hybrid assay, AtMKK4 interacts strongly only with AtMPK3/6, and AtMKK5 interacts only with AtMPK6 (Lee et al., 2008), but in the protein chip analysis, AtMKK4 was found to phosphorylate AtMPK3, AtMPK5, AtMPK6 and AtMPK10, while AtMKK5 seemed to be more promiscuous, phosphorylating AtMPK3, AtMPK4, At MPK5, AtMPK6, AtMPK8, AtMPK10 and AtMPK16 (Popescu et al., 2009). Thus, additional in vitro phosphorylation experiments on particular CmMKK and CmMPK interactions are needed.

Conclusion

This study is the first transcriptome-wide analysis of the MPK and MKK family in C. morifolium. The transcription patterns of 11 CmMPKs and six CmMKKs were investigated in different organs of chrysanthemum, as well as in response to various phytohormones and abiotic stresses. In addition, the possibility that CmMKK4-CmMPK13 and CmMKK2-CmMPK4 may be involved in regulating salt resistance and the relationship between CmMKK9-CmMPK6 and temperature stress are first mentioned in this study. These findings lay the foundation for future research on the functions of CmMPKs and CmMKKs in plant stress response and growth, which will promote their application in chrysanthemum breeding.

Supplemental Information

Supplemental Information 1 Dataset 1. Sequences used in this manuscript.

Click here for additional data file.

Supplemental Information 2 Table S1. Primers used in the manuscript.

Click here for additional data file.

Supplemental Information 3 Table S2. Interactions between MKK and MPK revealed by yeast two-hybridization.

Click here for additional data file.

Additional Information and Declarations

Competing Interests

Author Contributions

Data Availability

The authors declare that they have no competing interests.

Aiping Song conceived and designed the experiments, performed the experiments, analyzed the data, prepared figures and/or tables, authored or reviewed drafts of the paper, approved the final draft.

Yueheng Hu conceived and designed the experiments, performed the experiments, analyzed the data, prepared figures and/or tables, authored or reviewed drafts of the paper, approved the final draft.

Lian Ding analyzed the data, approved the final draft.

Xue Zhang performed the experiments, approved the final draft.

Peiling Li performed the experiments, approved the final draft.

Ye Liu contributed reagents/materials/analysis tools, approved the final draft.

Fadi Chen contributed reagents/materials/analysis tools, authored or reviewed drafts of the paper, approved the final draft.

The following information was supplied regarding data availability:

The isolated 11 CmMPK (GenBank: MG334201–MG334212) and 6 CmMKK (GenBank: MG334196–MG334201) sequences are accessible via GenBank.

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
