# Peer review of "Comprehensive analysis of mitogen-activated protein kinase cascades in chrysanthemum"

_PeerJ, doi:10.7717/peerj.5037_

## Round 0.1 · original submission · Minor Revisions

As you can see, both reviewers provide very favorable comments and think that your manuscript is potentially publishable. However, they also indicated that there are some minor issues that need to be addressed. Therefore, please carefully consider all critiques raised by both reviewers and revise your manuscript accordingly.

Reviewer 1 ·

Basic reporting

In this manuscript, the authors conduct a good study on MAPK cascades of chrysanthemum. With gene sequencing and PCR, a couple of CmMPK and CmMKK genes are characterized and expression pattern of these genes are quantified. In addition, the interaction between CmMMK and CmMPK are unveiled with yeast two-hybrid assay and their contribution to stress resistance of chrysanthemum are evaluated in various stressed growth conditions.

The author gains a huge amount of experiment data with solid work as well as a comprehensive genetic analysis. The conclusions are clearly stated in the manuscript and reasonably made according to the experiment data. Yet, there are several issues in this manuscript such as loss of statistic test and incomplete figure legend, which can cause the confusion and misinterpretation. Overall, the manuscript is publishable after all the issues are fixed properly. Therefore, I suggest minor revision on this manuscript.

Major Issues
1. In figure 3&4, the author does NOT provide any statistic tests to show the significance of difference. To confirm this, statistical analysis needs to be added in the method part to explain like what statistical test is used, what p value is, etc.
2. For line 323, the CmMPK13 and CmMKK4 have opposite expression level in root according to figure 3a and 3b, (MPK 13 is low but MKK4 is high). Given root is the main organ responsive for osmotic stress, should this need to be involved in the discussion here? Also, it seems that the expression level data in figure 5 showing the similarity comes from the whole plant but not the root, so it is questionable that how relevant this data is to plant salt resistance. As shown in figure 3a and 3b, the expression of MPK13 and MKK4 in other part of plant (i.e. stem) may offset their difference in root, which leads to the overall similar expression.

Minor Issues
1. Reorganize the abstract part following the instruction of Peer J. No need to highlight the background, method etc..
2. In the abstract and line 20, the sentence that “qRT-PCR and yeast two-hybrid assay were used for the further analysis” is ambiguous, the author should clearly state what analysis is done by qPCR and yeast assay.
3. Beside the figure, author should also state what type of plant each sign (blue circle, green square and yellow triangle) represent in the figure legend.
4. In result 3.1, the author should be clear on what criteria or standards to divide all the motifs into four groups for the comparison. If they are grouped by software, then what logarithm does the software use should be mentioned.
5. In figure 4, what is CK, C4, and H40 stands for? Clarify them in the figure legend.
6. In line 224, what is the resistance that MPK3 plays role in?
7. In line 249, should be among not between the MKKs.
8. In line 317-319, figure 5 should be referred after “its expression was significantly increased after PEG and salt treatments” since the figure3 doesn’t have data to support that.

Experimental design

no comment

Validity of the findings

no comment

Additional comments

no comment

Reviewer 2 ·

Basic reporting

In their manuscript submitted to PeerJ, Aiping Song et al. present an interesting finding about the MKK and MPK genes from Chrysanthemum sp.. From phylogenetic and conserved motif analysis from different plant species, authors have characterized 6 MKK genes and 11 MPK genes from Chrysanthemum sp. and their response to phytohormones and stresses.
Introduction has been well-written with adequate references emphasizing on the important role of Mitogen Activated Protein kinases (MAPK) in plants and their crucial role in various signal cascade in response to stress or other physical requirements like growth, differentiation. The authors thus, successfully highlight the importance of 11 MPK and 6 MKK from Chrysanthemum sp and their interaction with each other in response to various signals forming different signal cascade finally emphasizing the novelty of this study. The manuscript has been drafted in accordance with PeerJ standards and all the figures are of high quality. However, for figure 4 author didn’t mention the meaning of the abbreviations CK, C4 & H40 in the figure legends which raises ambiguity in data interpretation. For each dot blot strip in figure 6, labeling of protein corresponding its dot is slightly misleading.

Experimental design

The authors through phylogenetic sequence analysis has identified 6 MKK genes and 11 MPK genes from Chrysanthemum sp.. Later, authors have functionally characterized these MKK and MPK genes using various experimental techniques. Experiments designed to address the unknown, are well defined and technically sounds good.

Validity of the findings

Authors reveal the important role 6 MKK genes and 11 MPK genes from Chrysanthemum sp. in response to plant growth, abiotic stress and phytohormones. The results/discussion as reported by authors need some edits as suggested below:
Authors for figure 4 didn’t mention the meaning of the abbreviations CK, C4 & H40 in the figure legends so it was difficult to interpret the results for section 3.3
The discussion is lengthy. I would appreciate if the authors make it more concise and interesting by including figure and less text.

---

## Round 0.2 · accepted · Accept

Since all critical points of both reviewers were adequately addressed and the manuscript was revised accordingly, I am pleased to accept it for publication in PeerJ.

#